# Critical Overview on Endocrine Disruptors in Diabetes Mellitus

**DOI:** 10.3390/ijms24054537

**Published:** 2023-02-25

**Authors:** Charlotte Hinault, Philippe Caroli-Bosc, Frédéric Bost, Nicolas Chevalier

**Affiliations:** 1Université Côte d’Azur, INSERM U1065, C3M, Bâtiment Universitaire Archimed, 151 Route de Saint-Antoine de Ginestière, BP 2 3194, CEDEX 3, 06204 Nice, France; 2Équipe Labelisée Ligue Nationale Contre le Cancer; 3Université Côte d’Azur, CHU, INSERM U1065, C3M, 06202 Nice, France

**Keywords:** endocrine-disrupting chemicals, insulin, metabolic disruptor, pollutant

## Abstract

Diabetes mellitus is a major public health problem in all countries due to its high human and economic burden. Major metabolic alterations are associated with the chronic hyperglycemia that characterizes diabetes and causes devastating complications, including retinopathy, kidney failure, coronary disease and increased cardiovascular mortality. The most common form is type 2 diabetes (T2D) accounting for 90 to 95% of the cases. These chronic metabolic disorders are heterogeneous to which genetic factors contribute, but so do prenatal and postnatal life environmental factors including a sedentary lifestyle, overweight, and obesity. However, these classical risk factors alone cannot explain the rapid evolution of the prevalence of T2D and the high prevalence of type 1 diabetes in particular areas. Among environmental factors, we are in fact exposed to a growing amount of chemical molecules produced by our industries or by our way of life. In this narrative review, we aim to give a critical overview of the role of these pollutants that can interfere with our endocrine system, the so-called endocrine-disrupting chemicals (EDCs), in the pathophysiology of diabetes and metabolic disorders.

## 1. Introduction

Diabetes mellitus is a group of metabolic disorders that is defined phenotypically by chronic elevations in blood glucose (i.e., hyperglycemia). According to the classical definition, it results mainly either from a deficiency in insulin secretion (type 1 diabetes mellitus [T1D]) and/or from a defect in insulin action, namely insulin resistance affecting liver and peripheral tissues (type 2 diabetes mellitus [T2D]). Today the distinction between T1D and T2D is equivocal, and the classification has been revisited with many other specific diabetes subtypes. The current prevalence of diabetes is unprecedented. Ten years ago, the World Health Organization (WHO) reported that 422 million people suffered from diabetes all over the world (95% of them with type 2 diabetes) [1], whereas the same organization predicted earlier in the 2000s that 330 million people would have diabetes in 2030. The latest report of the IDF (International Diabetes Federation) estimated this number as 537 million people in the world, representing 1 in 10 adults (20–79 years) [2]. Moreover, across the world, 240 million more adult people (20–79 years) have impaired glucose tolerance, which is a predictor for the development of diabetes [2]. Importantly, due to the chronic complications arising from uncontrolled chronic hyperglycemia, diabetes imposes significant economic burdens across the world. The IDF estimated that 10% of global health expenditures are spent on diabetes, with a total annual economic cost estimated to more than USD 760 billion in the world, including a notable annual cost of USD 330 billion in the United States for example [3].

The reasons for this rapid increase in diabetes remain unclear. At the same time, the prevalence of obesity has increased in the same proportion across the world. Excess caloric consumption and a sedentary lifestyle are undoubtedly key causal factors for obesity and diabetes. However, there is growing interest in the contribution of “non-traditional” risk factors such as environmental chemicals, micronutrients, or gut microbiome to the etiology of these health conditions. Indeed, concurrently with the escalation of diabetes prevalence, the world has witnessed the massive production and release of toxicants into the environment. Many of these can interfere with the endocrine system by altering the production, release, transport, and action of hormones; these are termed endocrine-disrupting chemicals (EDCs). Since the beginning of the 2010s, a growing body of both in vitro and in vivo evidence suggests that EDCs play a significant role in the etiology of diabetes and metabolic disorders. The US National Institute of Environmental Health Sciences (NIEHS) concluded that the existing literature provided plausibility, varying from suggestive to strong, that exposure to environmental chemicals may contribute to the actual epidemic of diabetes [4]. The present review will give a critical overview of the role of EDCs in the etiology of diabetes, by discussing available epidemiological data concerning T1D and T2D, experimental data demonstrating links between chemical exposure and changes in insulin action or secretion, and by highlighting the role of EDCs on metabolic fetal programming. We systematically searched animal and human studies available up to 15 September 2022. We conducted a literature search in Scopus, ISI Web of Science and PubMed using the following keywords to identify relevant articles: “endocrine disruptors”, “chemicals”, “pollutants”, “metabolic syndrome”, “insulin resistance”, “glucose intolerance”, and “diabetes”.

## 2. Human Implications of Diabetogenic Pollutants

### 2.1. Persistent EDCs and Type 2 Diabetes

The most representative EDCs showing a well-known causal relationship with obesity and diabetes are persistent organic pollutants (POPs), which contain two main groups of synthetic compounds: polycyclic aromatic hydrocarbons and halogenated hydrocarbons. All the known POPs are listed in the Stockholm Convention (available at http://chm.pops.int/TheConvention/ThePOPs/AllPOPs/tabid/2509/Default.aspx (accessed on, 10 February 2023)). Among these, the best known are organochlorine (OCs) pesticides (dichloro-diphenyl-trichloroethane (DDT) and its metabolites chlordane) and industrial chemicals (polychlorinated biphenyls (PCBs), polybrominated diphenyl ethers (PBDE), dioxins (TCDD: 2,3,7,8-tétrachlorodibenzo-p-dioxine), and per- and polyfluoroalkyl substances (PFAs)). These anthropic chemicals are resistant to degradation, possess low solubility in water but high solubility in lipids and tend to accumulate in the adipose tissue with the highest concentrations in humans. They have long half-lives (exceeding decades in the environment) and are not well metabolized or excreted, explaining why even small amount that are daily ingested can accumulate to yield detectable amounts over time [5]. They were involved in several main health concerns, including cancer [6], damage to neurological and reproductive systems [7], but the awareness of their metabolic concern was the result of large-scale casualties demonstrated by mass poisoning incidents.

Indeed, a chemical plant explosion in Seveso, Italy, was responsible for a massive and acute release of dioxins in the summer of 1976. Several years after this environmental disaster, follow-up studies showed an increased risk of T2D in exposed people, especially in women [8]. Interestingly, this risk was higher in the youngest women, with a significant increased risk in women aged 12 and under at the explosion (adjusted odd ratio [OR] for metabolic syndrome in women = 1.05 [95% CI 0.78–1.43]; before 12 years, adjusted OR = 2.03 [95% CI 1.25–3.30]; above 12 years, adjusted OR = 0.96; [95% CI 0.68–1.35]; *p* interaction = 0.01) [9]. Similarly, the study of military troops exposed to dioxins contained in the “Agent Orange” herbicide used by the US Army Corps, Operation Ranch Hand, during the Vietnam War, revealed the same association between serum TCDD concentration and the prevalence of T2D (relative risk [RR] = 1.5; [95% CI 1.2–2.0]) [10]. The consumption of adulterated rice bran cooking oil (“Yusho” oil) in a group of Taiwanese, also known as the Yu-cheng incident, was related to a higher prevalence in T2D, especially in women (OR = 2.1; [95% CI 1.1–4.5]), due to a contamination with PCB ethers, dioxins and furans [11]. A cross-sectional study of Taiwanese living in a highly dioxin-contaminated area also reported a strong correlation between abdominal obesity and insulin resistance (determined by the HOMA index) in people with the highest serum levels of dioxins (OR = 5.23; [95% CI 3.53–7.77] in men; OR = 4.57; [95% CI 2.70–7.64] in women) [12]. However, those poisoning events are not characteristic of most environmental exposures.

To document the role of pollutants in diabetes burden, authors have mainly reported occupational case-control cohorts that in fact exhibit many limitations: the exposure occurs many years before the analysis is undertaken and the levels of EDCs are usually estimated by questionnaires or back calculation from serum or urine concentrations measured many years later. For example, in the D.E.S.I.R. cohort, incident T2D was not associated with POPs levels measured in plasma nine years before the T2D diagnosis [13]. In addition, most of these occupational studies focused on exposure to one EDC at a time, while the cohorts are usually exposed to a whole mixture of toxicants and studies usually focus on individuals who live in heavily polluted areas [14]. Levels of PCBs were associated with higher fasting plasma glucose and an increased risk of T2D in a Chinese population (OR = 1.558 when fourth quartile [Q4] was compared to first quartile [Q1] for PCB52 [95% CI 1.109–2.189; *p* = 0.025]; OR = 1.841 Q4 vs. Q1 for PCB153 [95% CI 1.275–2.656; *p* = 0.001]) [15]. In a cohort of obese adolescents and young adults (SOLAR and CHS cohorts), PCB118 and PBDE153 were associated with higher glucose concentrations in oral glucose tolerance tests [16]. Another critical point to consider is the exposure dose associated with the occurrence of T2D. Generally, people are exposed to low to very low doses of EDCs that are usually at least hundred times lower than no or low adverse effect levels according to regulatory guidelines [17]. However, these low to very low doses were related to incident T2D, especially in the case of exposure to PFAS, while higher doses were not related to adverse metabolic effects [18,19], suggesting a non-monotonic dose-response relationship following parabolic-shaped or inverted-U-shaped curves, as previously reported for bisphenol A [20].

Thus, all these results should be interpreted with caution and may not reflect the exact risk incurred by the general population. For this reason, the usual figures adapted from Neel et al. [21] and Baillie-Hamilton P. [22] illustrating the spectacular concordance between the respective evolution of diabetes and obesity prevalence and the national production of synthetic organic compounds in the United States should be carefully considered.

Finally, the association between POPs exposure and diabetes may be due to confounding factors such as fat mass, as most of them are stored in adipose tissue and/or as POPs are also considered as obesogens [23]. Thus, POPs exposure may have a synergistic effect with increased adipose tissue on the risk of T2D [24]. However, some studies reported that POPs were related to T2D only in lean patients but not in overweight or obese patients, suggesting a specific role of POPs in T2D pathophysiology [25], probably through modifications of insulin synthesis and/or insulin release [26].

### 2.2. Non-Persistent EDCs and Type 2 Diabetes

Since the beginning of the 2000s, several longitudinal studies have investigated the potential role of persistent and non-persistent EDCs, with a special concern for phthalates, especially bisphenol A (BPA), because more than 95% of all people worldwide have BPA in their bodies [27]. Indeed, BPA is probably one of the most extensively studied and well-known EDCs. First synthesized in 1891, BPA exhibits estrogenic properties that have aroused concern for human health, especially regarding developmental and reproductive effects [28], and cancers of hormone-dependent organs [29].

Among the population studies, the US National Health and Nutrition Examination Survey (NHANES), which is an American food consumption database program operating since 1960, is probably the most documented survey, monitoring 5000 to 10,000 representative Americans annually. Since 1999, it also includes the testing of an extensive number of chemicals and systematically reports detectable blood and/or urine levels of several chemicals in most participants. Moreover, diagnosed or self-reported diabetes was strongly associated with the exposure to POPs (PCBs, dioxins, *p*,*p*′-DDE) but also with non-persistent EDCs such as phthalates and BPA (OR = 2.74 [95%CI 1.44–5.23] after adjustment for age, gender, body mass index, and ethnicity) [30,31]. Other cross-sectional studies reported a suggestive to strong association between diabetes and pre-diabetes and EDCs exposure, especially for BPA exposure [32,33,34]. For example, Sun Q. et al. showed a positive correlation between urinary BPA and butyl-phthalate concentrations (assessed from blood and urine samples collected 5 to 10 years before) and the incident T2D in the participants of the Nurses’ Health Study II (NHSII), but not in the older NHS counterparts [35]. In a meta-analysis of all clinical studies on BPA published before mid-2014, urinary concentrations of BPA were associated with T2D risk (OR = 1.47 between highest and lowest urinary levels) [36]. This risk was confirmed by BPA CLARITY, a project begun by the NIEHS (National Institute of Environmental Health Sciences)/NTP (National Toxicology Project) and the FDA, which confirmed significant metabolic adverse effects at the lowest dose examined (2.5 μg/kg body weight per day), far lower than the maximum safe daily oral BPA dose over the lifetime (50 μg/kg body weight per day) established by the EPA and FDA (CLARITY-BPA Program (https://ntp.niehs.nih.gov/whatwestudy/topics/bpa/index.html?utm_source=direct&utm_medium=prod&utm_campaign=ntpgolinks&utm_term=clarity_bpa (accessed on 10 January 2023))).

The health concerns of BPA led to the progressive use of substitutes since about ten years ago. Urinary and blood levels of these substitutes (notably bisphenol S, F, M, B AP, AF, BADGE) are gradually rising in the global population. However, limited data are available concerning their innocuity for human health. One of the most investigated substitutes is bisphenol S (BPS) but only a few studies are available in the literature regarding the association of exposure to BPS with incident diabetes, fasting blood glucose, or insulin resistance [36,37,38,39,40]. Among them, two reported a significant association between exposure to BPS and incident or overt diabetes with variable odd ratios: OR 3.83 [95% CI 2.37–6.20; *p* < 0.001] [37] and adjusted hazard ratio (aHR) at year 3:2.81 [95% CI 1.74–4.53] [36]. As for BPA, this increased risk is not dependent on overweight or obesity, as urinary BPS concentrations were positively associated with an increase in insulin resistance evaluated by ln-transformed homeostatic model assessment for insulin resistance (ln-HOMA-IR) (*p* = 0.017) only in women not overweight or obese (BMI < 25 kg/m^2^) [41].

Phthalates are plasticizers with well-known EDC properties, especially in obesity determining. Recently, the evidence for a role in the determining of diabetes has been mixed. Indeed, some reported a positive correlation between phthalates exposure and insulin resistance determined by the HOMA index [42]. Moreover, in a meta-analysis of seven studies including more than 12,000 people, exposure to low and high molecular weight phthalate metabolites, but also to di-2-ethylhexyl phthalate (DEHP), was significantly associated with an increased risk of incident or overt type 2 diabetes, with an overall OR of 1.69 [95% CI 1.29–2.20], 1.71 [95% CI 1.43–2.05] and 2.15 [95% CI 1.48–3.13], respectively [43].

Other EDCs have been suggested in the diabetes burden. Triclosan and triclocarban are two biocides commonly used despite an FDA ban in 2016, especially in toothpastes. As for phthalates, evidence in human studies has been mixed. In a recent analysis of NHANES (2013–2014), a significant positive association was reported between exposure to triclocarban and incident or overt diabetes, but only in women (OR = 1.79 [95% CI 1.05–2.05] in women vs. 1.29 [95% CI 0.73–2.27] in men; *p* = 0.032), even after adjusting for potential confounding factors [44]. Heavy metals (cadmium, lead, and mercury), trace elements (such as zinc), and arsenic have also been suggested as risk factors for metabolic syndromes or overt diabetes. However, results remain conflicting [45,46,47,48,49]. More recently, some authors reported a possible role of the air pollution in the diabetes burden, including ozone [50], nitrogen dioxide [51], and particulate matter (PM), especially PM2.5, which were related to an increased incidence and prevalence of diabetes [52] and worsened glycemic control among people previously diagnosed with diabetes [53]. It is important to note that the impact of air pollution is enhanced by warm and humid weather, which raises important concerns about the possible impact of climate change on diabetes risk [54].

However, several limitations with these epidemiological studies should be highlighted. As in any observational study, the association between EDCs exposure and diabetes does not establish causality. Furthermore, many of the above studies were designed to correlate diabetes prevalence with current EDCs levels, while their current levels may strongly differ from concentrations during the disease development. Moreover, EDCs are usually highly correlated with each other, and it is not always possible to precisely determine the independent effect of each chemical compound when people are daily exposed to a whole mixture. Another point to consider is the potential individual differences in EDCs metabolism. Indeed, genetic polymorphisms could affect both bioavailability and adverse effects of EDCs, notably during critical period of exposure, such as in pregnancy and early childhood, as it was notably reported for BPA [55]. Finally, the association between EDCs exposure and diabetes may be due to confounding factors, as suggested above [13,56,57].

### 2.3. EDCs and Type 1 Diabetes

The link between environmental toxicant exposures and T1D has been relatively understudied compared to T2D. However, the prevalence of T1D is also increasing to a lesser extent but with notable geographic variations, especially in industrializing countries such as China, for example. Several cross-sectional studies reported conflicting results for EDCs: some studies were characterized by higher concentrations of EDCs at diagnosis of T1D in children [58], while other suggested a protective effect of some EDCs (especially perfluoroalkyls) on the development of T1D in children [59]. As T1D arises from autoimmune destruction of pancreatic beta cells, it has been suggested that EDCs could be involved by promoting beta cells apoptosis or necrosis and/or by disrupting the immune system (notably for BPA and its substitutes) [58]. Other epidemiological studies focused on the role of air pollution on T1D incidence and reported that exposure to air pollution (notably to ozone) during pregnancy is associated with an increased risk for early T1D in children (HR per interquartile increase = 2.00 [95% CI 1.04–3.86]) [60,61]. Those data were recently confirmed in an environment-wide association study conducted in the UK on about 14,000 children with T1D, which reported a spatial heterogeneity in disease mapping depending on 15 main risk factors including air pollutants (PM_2_._5_, nitrogen dioxide), lead, radon, outdoor light at night, overcrowding, population density, and ethnicity [62]. Overall, more work is needed to better understand the contribution of EDCs and environment to T1D risk.

### 2.4. Maternal Exposure to EDCs and Diabetes

Another critical issue to consider is exposure to EDCs during pregnancy, which opens two lines of research, one in relation to the development of diabetes during pregnancy and the other one in relation to the development of diabetes in the offspring [63]. Indeed, the fetal period is a critical window, which can definitively influence the developmental origins of health and disease (DOHaD) through modifications in gene expression and unexpected effects on metabolism of progeny as proposed by David Barker [64]. Therefore, either gestational diabetes potentially induced by EDCs or the EDC exposure during this critical window may have an impact on progeny predisposition to develop diabetes. The data from human studies investigating the potential impact of prenatal exposure to EDCs on mother metabolic dysfunction are equivocal. Some studies have shown that exposure to phthalates and/or BPA was associated with increase plasma glucose and impaired glucose tolerance in pregnant women [65]. The same results were recently reported for BPS, which impaired blood glucose tolerance during pregnancy but was not associated with incident gestational diabetes [40]. Yan et al. have recently published a meta-analysis of 25 studies including 16 cohort studies, 7 case-control studies and 2 cross-sectional studies to conclude that PCBs, PBDE, PFAs and phthalates are critical gestational diabetes risk factors [66]. Other epidemiological studies have found associations between heavy metals [67,68], PFASs, POPs [69,70], or some bisphenol substitutes (notably bisphenol AF) [40] and an increased risk of gestational diabetes. Overall, there is some evidence for the role of EDCs in gestational diabetes, but further studies are needed to better characterize this risk.

Importantly, several studies have shown that mothers can directly expose their progeny to EDCs, for instance BPA, through the placenta but also by breast feeding [71,72]. Others have suggested associations between prenatal exposure to DDE and PFAs and predisposition to develop diabetes into progeny [73]. Moreover, prenatal exposure to pollutants is associated with an increased incidence of overweight and obesity in childhood and adolescence, with early catch-up growth as previously described by David Barker [64] in the DOHaD hypothesis. For example, in the CHAMACOS (Center for the Health Assessment of Mothers and Children of Salinas) study, prenatal exposure to mixtures of phthalates and phenols [74,75] or to DDT and its metabolites (i.e., DDE) [76] was associated with increased risks of overweight and obesity at the age of 5 and 12 years for both boys and girls. In this cohort, the risk of T2D in progeny has not yet been assessed but we can assume that such early metabolic disorders may lead to an increased prevalence of T2D.

## 3. In Vivo Evidence of Diabetogenic Pollutants

Epidemiological evidence is necessary to evaluate the potential impact of EDCs on human health, but although usually correlative, studies have not been sufficient to establish that a molecule is an EDC. In vivo and in vitro experiments are required to identify the doses, the kinetics, and the molecular mechanisms of the interferences with the endocrine systems even though they do not exactly replicate human physiology and variability/heterogeneity [77,78,79,80]. Using phenotypic and functional analyses in addition to complementary omics approaches, as carried out on humans [81], animal experiments specifically can study pathophysiological features from periconception to adult life up to progeny. Of note, some studies have been performed and well-reviewed for EDCs as obesogenic pollutants, showing their impact on adipose tissue hyperplasia and/or hypertrophy [23,78,82,83]. However, obesity is not the only cause of T2D and is not necessarily linked to diabetes. EDCs have also been implicated as diabetogenic pollutants as previously reviewed [56,82,84,85,86,87,88]. To show diabetogenic effects of EDCs in vivo experiments usually measure insulin secretion and glucose metabolism dysregulations due to insulin resistance. Of note, some studies have also started to explore the effects of EDCs on gut microbial ecology and physiology, and their subsequent impact on glucose metabolism as gut microbiota modifications have been associated with diabetes [89]. In vivo evidence is summarized in Figure 1.

### 3.1. Persistent EDCs and T2D

Global studies have been conducted through consumption of fat-containing food, considering that we are exposed to pollutants this way. By feeding rats [90] or mice [91] during a month with a high-fat diet along with natural source of a cocktail of persistent EDCs present in salmon oil, Ruzzin et al. were the first to show that these rats became insulin-resistant. This chronic exposure to low doses of POPs exacerbates obesity and hepatic steatosis with a dysregulation of key hepatic genes. When a similar type of diet, this time supplemented with a cocktail of commonly found EDCs (TCDD, PCB153, DEHP and BPA) instead of salmon oil, was maintained from periconception period throughout life in mice, this led to sex-specific metabolic alterations in the progeny [92]. A chronic exposure to a PCB mixture (Aroclor 1254) for 30 weeks led to hyperinsulinemia independent of body weight in both standard and high-fat diet-fed mice, while an aggravation of insulin resistance was observed in high-fat diet-fed mice [93]. Another study showed that a chronic exposure of this mixture every 3 days for 60 days increased body weight and insulinemia associated with insulin resistance development [94]. Even though a mixture is more representative of our everyday exposure, we are not exposed to the same cocktails and not in the same proportions. Therefore, animals were also exposed to EDCs one by one rather than in cocktails to identify the effects of each EDC to avoid potential synergic or antagonist activities of one EDC on the others. Regarding PCBs, a 2-week exposure either with PCB126 or PCB77, two coplanar PCBs, was sufficient to provoke an impairment of glucose tolerance associated with insulin resistance in standard diet-fed mice [95]. Similar data were obtained with chronic exposure of PCB-118 or PCB-138 in standard diet-fed mice [96], while for PCB153, a deterioration of glucose and insulin intolerance in high-fat diet-fed mice was shown [97]. Interestingly, PCB126 had a strong effect in inducing hepatic steatosis in mice compared to a PCB mixture (Aroclor 1260); conversely, the mixture had the strongest deleterious effect on pancreatic beta cell function. However, in both cases, insulin resistance was not measured [98]. Concerning TCCD, an acute exposure has been shown to impair glucose-stimulated insulin secretion without hyperglycemia in WT mice but not in AhR (Aryl hydrocarbon Receptor)-deficient mice, suggesting that this effect is mediated though this nuclear receptor [99]. A single exposure to TCDD can also abolish basal insulin levels up to six weeks post-injection in both sexes, but adverse effects on glucose homeostasis are sex-dependent, as they were observed only in males [100]. Moreover, chronic exposure to TCDD may induce insulin resistance in rats [101].

Adult exposure to other persistent EDCs such as DDT, and its metabolite DDE, or TBT were shown to have obesogenic effects, and subsequently potential diabetes risk factors rather than to have direct diabetogenic effects [23,82]. All these studies show that EDCs impact on glucose homeostasis and insulin secretion, but the phenotypes have a variable intensity dependent on the doses but also the exposure reflecting the complexity of the problem [102,103].

### 3.2. Non-Persistent EDCs and T2D

Studies of BPA exposure and the development of T2D in rodent data can appear contradictory because of different methods, the windows, the exposure, and the doses, but they provide convincing evidence of adverse effects of BPA contributing to the development of diabetes. Several reviews have recently outlined key publications studying mechanisms linking BPA and T2D development [55,71,82,104,105,106]. In summary, it is now clear that BPA exposure leads to perturbated insulin secretion with β cell dysfunction, liver and muscle insulin resistance but also increased adipogenesis. Recent studies have investigated the metabolic outcomes of chronic and prenatal exposure to BPA substitutes in rodents. Despite some conflicting results (notably between males and females, which seem less susceptible), BPA substitutes, in particular bisphenol S (BPS), exhibit the same metabolic adverse outcomes than BPA, involving similar signaling pathways, including especially PPARγ and extranuclear-initiated ER pathways [107,108,109].

Contrary to BPA, few studies have shown in vivo that phthalates/DEHP are linked to diabetes interfering with insulin secretion [110,111]. Phthalates have rather been linked to insulin resistance and obesity, notably increasing oxidative stress. Interestingly, Wei et al. have shown that chronic exposure of mice to DEHP deregulates oxidative stress in muscle though the miR-17/Keap1-Nrf2/miR-200a axis leading to insulin resistance [112]. Recently, Baralic et al. highlighted a potential additive effect for a mixture of DEHP, DBP and BPA regulating redox status parameters such as TOS, SOD, and SH groups in rat pancreatic tissue [113].

Importantly, as reviewed by Castriota et al. [114], recent studies in rodents have shown that arsenic exposure in adult life induced glucose intolerance and decreased glucose-stimulated insulin secretion in mice [115], and led to impaired glucose sensing and subsequently insulin release in rats [116]. In addition, chronic exposure to arsenic in drinking water have been shown to impact body weight and composition, fasting glycemia, and insulinemia [117]. These data have shown an increased HOMA index in several different outbred mice [117].

### 3.3. EDCs and T1D

Concerning the potential causal relation between EDC and T1D, some studies have been carried out mainly in non-obese diabetic (NOD) mice [118]. So far, no consensus has been obtained as too few studies have been performed, and because both immunosuppressive and immunoproliferative effects have been reported. Regarding PCBs, Kuiper et al. have shown that, either at high or low doses in short or chronic exposures, the development of T1D was decreased in NOD mice, associated with a decrease in T cell populations [119]. These data were obtained in the context of conflicting epidemiological studies [120,121,122], and there have been no further studies to corroborate. Likewise, for PFAs only one published work in NOD mice showed accelerated insulitis [123], and for inconsistent epidemiological studies regarding this, see [118]. While NOD mice exposed to DDE displayed increased incidence of T1D [124], contradictory results were previously obtained on the immune response in mice after DDT exposure [125]. The situation is not clearer for dioxins as TCDD. Indeed, the exposure of WT mice leads to impaired-second-phase of glucose-stimulated secretion [99], whereas the exposure of NOD mice reduced insulitis through the increase in Foxp3 T cells [126]. In both cases, the effect of TCDD has been shown to be mediated through AhR. Concerning non-persistent EDCs, the same observation can be made since there are so far not enough studies. Exposure to BPA, but not to phthalates or to mixed BPA and phthalates, accelerated the development of T1D in NOD mice [127], yet the developmental exposure to phthalates led to pancreatic beta cell dysfunction with impaired glucose tolerance in rats [58,110]. Importantly, for BPA, but also BPS, effects on T1D development depend on the exposure window, the diet, and the sex [128,129]. However, regarding BPA, data remain controversial. Few data were also obtained in NOD and STZ mice after exposure to arsenic and air pollution [118], reinforcing the conclusion that further in vivo experiments are required to obtain concrete evidence for the adverse impact of EDCs on autoimmune responses and the development of T1D. EDCs have mainly been investigated for immunomodulation effects and some studies have measured their impact on gut microbiota or vitamin D levels as other mechanisms involved in the development of T1D [89,118]. However, further experiments notably in vitro are required to clarify their action and to identify their precise action modes at the molecular level.

### 3.4. Maternal-Fetal Exposure to EDCs and Diabetes

While epidemiological studies focus on the potential impact of EDCs on the development of gestational diabetes, animal studies with EDC exposure during pregnancy usually investigate the effect on the development of diabetes in the offspring. Not many studies have been performed to study the effects of EDC developmental exposure. Interestingly, the rat dam exposure from day 8 to 15 with DDE led to transgenerational glucose intolerance and pancreatic impairment in F1 and F2 associated with modified genomic imprinting on igf2/H19 gene expression transmitted to F3 progeny through the male germ line [130,131]. Regarding BPA however, data remain contradictory [65,105,132]. Several studies have shown that developmental exposure to BPA leads to transgenerational abnormalities, but the degree of the phenotypes associated in progeny varies in terms of T1D or T2D development [105,129,133,134]. These conclusions are similar for other EDCs, with even counteracting effects when used as mixture.

## 4. In Vitro Evidence of Diabetogenic Pollutants

To complete animal experimental data, in vitro approaches are required to better decipher and identify the action modes of EDCs, but also to bring robustness in data with the possibility to work with complementary cellular models and in various species. Different approaches have been developed to set up in vitro screenings for EDCs, for instance with 96-well cell plates with isolated mice islets using ROS production measurement as a read out for damage in islet cells [135] or with human embryonic stem cell culture to study developmental origins of diabetes [136]. Other in vitro assays have been developed to study the impact of these EDCs on beta cell function [103,137]. However, the question is more than a screen and requires identifying how or at which level of the hormonal pathway EDCs act. Multiple mechanisms have been implicated for EDC interference action modes: through nuclear receptor hormone pathways (ER/AhR/PPAR) resulting in genomic effects, but also through insulin-signaling pathways (PI3K/ERK/AKT) implicating both genomic and non-genomic effects [82,87,88]. In both cases, they can induce or exacerbate an oxidative stress, a dysregulation of KATP channel or interference with other pathways involved in insulin exocytosis in pancreatic islets or adipose tissue inflammation and/or hepatic metabolism dysregulation [56,82,86,87]. Classically, among the main studies EDCs, PCB, PFOA and TCDD have been shown to interfere with PPARs, TCDD is also able to bind ER and AhR, and DDT with RXR /CAR. In vitro pathways involved in T2D are summarized in Figure 2.

### 4.1. Persistent EDCs and T2D

Concerning insulin secretion, an acute exposure to a PCB mixture (Aroclor 1254) was initially shown to stimulate glucose-induced insulin secretion of RINm5F cells potentially through calcium and CaM kinase [138,139]. Lee et al. showed that high concentrations of this same mixture decreased insulin release in Ins1E cells, while low concentrations had a stimulatory effect [140]. Ex-vivo analyses of tissues from mice chronically exposed to Aroclor 1254 showed an increased pancreatic beta cell mass associated with a decreased expression of key actors in the insulin-signaling pathway (IR/IRS/PI3K/AKT/GLUT4) in liver and muscle [94]. Interestingly, Kim et al. showed that in 3T3-L1 adipocytes and mice adipose tissue, the lipid droplet-associated protein, fat-specific protein 27 (Fsp27), mediated PCB-induced insulin resistance for both PCB118 (a dioxin-like PCB) and PCB138 (a non-dioxin-like PCB) through the downregulation of IRS1 associated with a phosphorylation decrease in PI3K and AKT [96].

Besides PCB, acute exposure of skeletal muscle (L6 cells) and pancreatic beta cells (Rin-m5F) to low doses of DDT (1 to 5 µM) impairs insulin-stimulated glucose uptake and insulin secretion, respectively, through increased production of intracellular reactive oxygen species (ROS) [141]. These subtoxic doses of DDT causes increased oxidative stress-induced activation of redox-sensitive kinases responsible for the impairment of insulin signaling with the decreased phosphorylation of IRS1/AKT [142]. Moreover, a proteomic approach revealed several markers of toxicity for acute but high doses of DDT (150 µM) in pancreatic beta cells, including oxidative stress proteins (GRP78, and endoplasmin), mitochondrial proteins (GRP75, ECHM, IDH3A, NDUS1, and NDUS3), proteins involved in the maintenance of the cell morphology (EFHD2, TCPA, NDRG1, and ezrin), and other proteins (vimentin, PBDC1, EF2, PCNA, and HSP27) [143]. Recently, it was reported that *p*,*p*′-DDE (10 µM) could also play a role in beta cell dysfunction by altering insulin translation, probably due to an increase in prohormone convertase levels [103,144] whereas low doses of TBT (0.1 µM) led to an increase in calcium-dependent insulin secretion through PKC and ERC phosphorylation [102,103].

Concerning dioxin, TCDD has been identified as an estrogen mimetic, which is a key regulator of glucose homeostasis by promoting energy homeostasis and improving insulin resistance, and beta cell function and survival [56,145,146]. Moreover, an acute exposure of pancreatic beta cells (Ins-1E) to nanomolar doses of TCDD has been shown to induce calcium influx involved in lysosomal and secretory granule exocytosis [147]. Kim et al. postulated that TCDD might exert adverse effects on beta cells inducing beta cell failure due to permanent insulin release. Inversely, morphological and functional alterations of pancreatic-isolated islets have been observed after acute exposure to nanomolar TCDD specifically in beta cells associated with decreased glucose-induced insulin secretion and dysregulation of key beta cell gene expression [148]. Besides diabetogen, TCDD has been identified as an obesogen interfering with insulin signaling (GLUT-4 [149]; TNF-a [150,151,152]) and adipocyte differentiation [80,153].

A perfluoroalkyls effect appears controversial in islets, for instance, for acute exposure in beta cells, while Qin et al. showed that PFOS stimulated calcium-induced insulin secretion through the activation of G protein-coupled receptor 40 [154]), He et al. showed that PFOA decreased insulin secretion through ER stress and the ATF4/CHOP/TRIB3 pathway [155] in hepatocytes [156]. Concerning effect in the liver, data are more consistent for involvement in steatosis [157,158,159].

### 4.2. Non-Persistent EDCs and T2D

Several reviews have largely discussed the link between BPA and diabetes covering the key studies bringing mechanistic and functional evidence reinforcing epidemiologic arguments [55,71,86,105]. BPA can act through several physiological receptors involved in glucose metabolism and insulin secretion, including nuclear receptors such as ER and PPAR but also membrane receptors such as GPR30 (membrane-bound estrogen receptors) or estrogen-related receptor (ERR) gamma. Recently, Park et al. showed that mixtures of BPA and its analog BPS and BPF have estrogen agonist and anti-androgen activities at lower doses than each one alone, highlighting a confounder variable in epidemiological approaches [160,161].

Concerning phthalates, in addition to activating oxidative stress in muscles [112], an acute exposure of pancreatic beta cells (Rin-m5F) to micromolar doses of phthalate induced a ROS-mediated PI3K/AKT/Bcl-2 pathway responsible for pancreatic beta cell apoptosis [162].

Regarding arsenic, a recent review robustly highlighted the potential molecular mechanisms mediating arsenic’s effects on glucose metabolism inhibiting SIRT3, AKT, and GLUT4 translocation [114]. Arsenic may interfere with calcium-mediated signaling necessary for insulin secretory granule exocytosis. Castriota et al. identified 16 genes commonly affected by arsenic, insulin resistance, and T2D (including insulin, IRS, adiponectin, and leptin) [114]. Arsenic has also been shown to modulate NLRP3 inflammasome activation involved in hepatic insulin resistance [163], or to be a master regulator of microRNA involved in pancreatic beta cell impairment [164]). Interestingly from a biophysical side, mercury and cadmium have the ability to modulate in a different manner the aggregation of human islet amyloid polypeptide (hIAPP) [165]. Cadmium promotes insoluble amyloid aggregation potentially leading to beta cell death, while mercury increases the formation of ion channel as an oligomer possibly involved in beta cell dysfunction [165].

Finally, many EDCs also exist in air as volatile or semi-volatile compounds in the gas phase or attached to PM. Hill et al. have recently demonstrated that chronic incubation of aortas from naive mice with plasma isolated from mice exposed to four times the recommended values of PM2.5 leads to a dyslipidemic phenotype, which could contribute to vascular inflammation and loss of insulin sensitivity [166].

## 5. Conclusions

Despite all these in vivo and in vitro studies, it remains difficult to form a comprehensive view on the causal relationship between EDCs and diabetes (resumed in Table 1). It is difficult to test, even for one molecule, different concentrations with various kinetics on different signaling targets. Even focusing on one pathway, we need thereby to multiply in vivo and in vitro approaches to obtain a complete analysis and we need to consider critical windows of exposition. Therefore, to definitely prove the role of EDCs in diabetes pathophysiology, further experiments are required. This would follow an experimental consensus with both acute and chronic exposures reflecting our daily exposure in accordance with observations from epidemiological data.

## Figures and Tables

**Figure 1 ijms-24-04537-f001:**
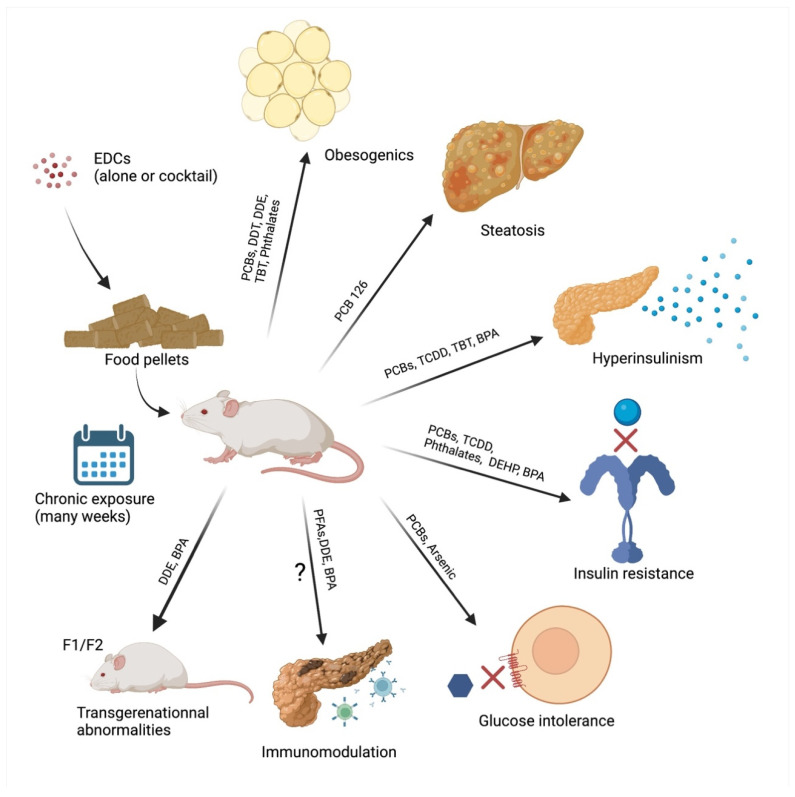
Schematic representation of the main EDC in vivo effects related to diabetes occurrence (BPA: Bisphenol A; DEHP: di-2-ethylhexyl phthalate; DDE: dichlorodiphenyldichloroethylene; DDT: dichlorodiphenyltrichloroethane; EDCs: Endocrine-Disrupting Chemicals; F1/F2: Filial generation 1/2; PCB: polychlorinated biphenyls; TBT: Tributyltin; TCDD: 2,3,7,8-Tétrachlorodibenzo-p-dioxine).

**Figure 2 ijms-24-04537-f002:**
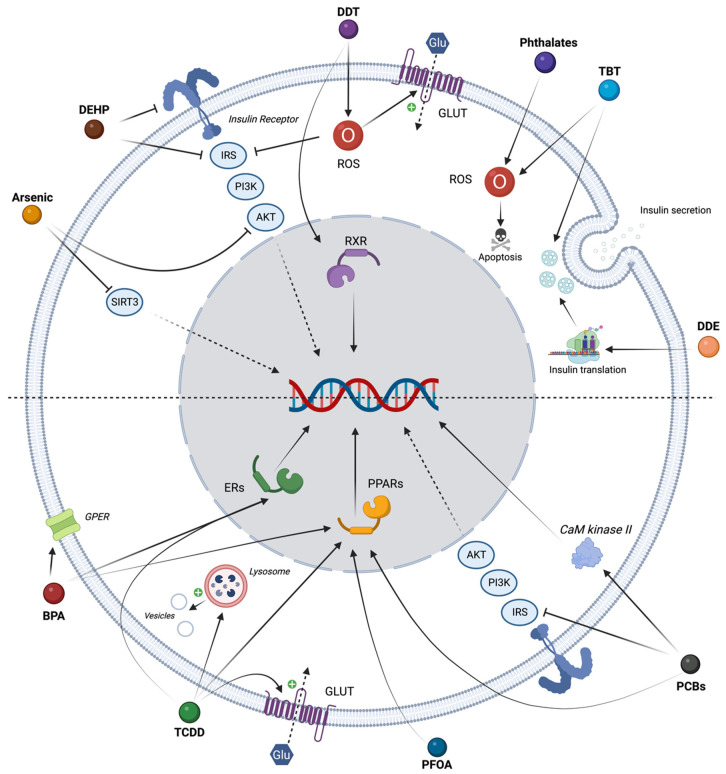
Schematic representation of the main EDC modes of action in relation to diabetes occurrence in adipocytes, hepatocytes or pancreatic beta cells. The figure summarizes the biological action in adipocytes, hepatocytes, or pancreatic beta cells. (AKT: Protein Kinase B; GPER: G Protein-coupled Estrogen Receptor; IRS: Insulin Receptor; ER: Estrogen Receptor; GLUT: Glucose Transporter; PI3K: Phosphatidylinositol 3 Kinase; PPAR: Peroxisome Proliferator-Activated Receptor; ROS: Reactive Oxygen Species; RXR: Retinoic X Receptor; SIRT3: Sirtuin 3).

**Table 1 ijms-24-04537-t001:** **Evidence for EDCs as obesogenic and diabetogenic** (T1D: type 1 diabetes; T2D: type 2 diabetes; GDM: gestational diabetes mellitus).

	EDCs	In Humans	In Vivo	In Vitro
Non persistent	BPA	Strong (T2D)/Low (T1D)	Strong (T2D)/Controversial (F1/F2)	Strong
	BPA substitutes	Limited data	Limited data	Limited data
	Phthalates/DEHP	Controversial (T2D)/Strong (GDM)/Strong (Obesity)	Low (T2D)/Strong (Obesity)	Strong
	Arsenic	Low (T2D)	Strong (T2D)	Strong
Persistent	PCBs	Strong (T2D)/Suggestive (T1D)/Strong (GDM)	Strong (T2D)/Strong (Obesity)	Strong
	PFOA	Low (T2D)/Strong (GDM)	Low (T2D)	Low
	TCDD	Strong (T2D)	Strong (T2D)	Strong
	DDT/DDE	Strong (T2D)/Strong (Obesity)	Strong (F1/F2)/Strong (Obesity)	Strong
	TBT	Low (T2D, T1D)/Strong (Obesity)	Suggestive (T2D)/Strong (Obesity)	Low

## Data Availability

Not applicable.

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
