# Peer review of "Critical Overview on Endocrine Disruptors in Diabetes Mellitus"

_ijms, 2023, doi:10.3390/ijms24054537_

Round 1
Reviewer 1 Report
This is a nice and well-written review manuscript, with nice images and a comprehensive table that summarizes the evidence until now regarding EDCs as a risk factor for DM. Thank you for the opportunity to review it. Bellow few comments I believe are helpful to improve the manuscript.
Title and Keywords – I suggest including mellitus at the end of the title and suppressing “diabetes” from the keywords since became repetitive. To include mellitus is important because diabetes insipidus, despite being quite rare, can be secondary to many conditions and this may help other researchers to refine their search.
Line 30 – Is important to mention there are many other specific diabetes subtypes.
Line 50 – “endogenous endocrine system” sounds a bit redundant, and I believe just “endocrine system” would be fine. In contrast, if there is an “exogenous endocrine system” (maybe if we consider pheromones as an example), there is any evidence about ECDs interfering in this interindividual chemical communication?
Line 137 – I believe the phrase read better if an “and” was added before “cancers of hormone-dependent organs.” Also, use the hyphen between hormone-dependent organs.
Line 274 – Of note, some studies…
Line 313 – please rephrase this sentence, it sounds a bit convoluted and difficult to understand.
Line 344 – please rephrase the end of this sentence. It seems there´s a comma between “fasting glycemia” and “and insulinemia” but in this case, another comma could be welcome after “Insulinemia”. The fact is that as it is, the phrase is difficult to understand.
Line 384 – Regarding BPA however, data remain controversial.
Fig 2 - Despite a reader of the IJMS probably being familiar with many molecules’ names and their classic abbreviations, figure 2 would benefit from a short legend of the abbreviations used and not cited elsewhere (RXR, ER, PPAR, GPER, SIRT3).
Line 438 – There is a symbol after the references, probably a typo.
Line 452 – Controversial instead “contreversal”
Line 475 – (…adiponectin, and leptin).
Line 494 – This sentence is too long and became a bit convoluted in the middle. In the end, I think the objective is to characterize these EDCs effects in diabetes. In the way is wrote looks like all the work that needs to be done is to just characterize the molecules themselves.
Table 1 – I suggest including obesogenic also in the title – “Evidence for EDCs as obesogenic and diabetogenic (….)”
Author Response
First of all, we thank the reviewer for his/her precious comments. We improved our manuscript with all the remarks, as discussed below:
Q: Title and Keywords – I suggest including mellitus at the end of the title and suppressing “diabetes” from the keywords since became repetitive. To include mellitus is important because diabetes insipidus, despite being quite rare, can be secondary to many conditions and this may help other researchers to refine their search.
A: As sensibly noticed, we added the term "mellitus" to the title of our article
Q: Line 30 – Is important to mention there are many other specific diabetes subtypes.
A: we added some precisions on diabetes classification.
Q: Line 50 – “endogenous endocrine system” sounds a bit redundant, and I believe just “endocrine system” would be fine. In contrast, if there is an “exogenous endocrine system” (maybe if we consider pheromones as an example), there is any evidence about ECDs interfering in this interindividual chemical communication?
A: we removed the term "endogenous". In fact, to our knowledge, there is no evidence about EDCs and pheromones system
Q: Line 137 – I believe the phrase read better if an “and” was added before “cancers of hormone-dependent organs.” Also, use the hyphen between hormone-dependent organs.
A: we added an "and"
Q: Line 274 – Of note, some studies…
A: we performed the modification
Q: Line 313 – please rephrase this sentence, it sounds a bit convoluted and difficult to understand.
A: we rewrote the paragraph about TCDD
Q: Line 344 – please rephrase the end of this sentence. It seems there´s a comma between “fasting glycemia” and “and insulinemia” but in this case, another comma could be welcome after “Insulinemia”. The fact is that as it is, the phrase is difficult to understand.
A: we added a comma
Q: Line 384 – Regarding BPA however, data remain controversial.
A: we added this remark in the manuscript
Q: Fig 2 - Despite a reader of the IJMS probably being familiar with many molecules’ names and their classic abbreviations, figure 2 would benefit from a short legend of the abbreviations used and not cited elsewhere (RXR, ER, PPAR, GPER, SIRT3).
A: we added the abbreviations to the legend of the figure 2.
Q: Line 438 – There is a symbol after the references, probably a typo.
A: we removed this symbol, linked to a false reference
Q: Line 452 – Controversial instead “contreversal”
A: we corrected the term
Q: Line 475 – (…adiponectin, and leptin).
A: we added the comma and the term "and"
Q: Line 494 – This sentence is too long and became a bit convoluted in the middle. In the end, I think the objective is to characterize these EDCs effects in diabetes. In the way is wrote looks like all the work that needs to be done is to just characterize the molecules themselves.
A: we corrected the sentence in order to increase the comprehension of our purpose.
Q: Table 1 – I suggest including obesogenic also in the title – “Evidence for EDCs as obesogenic and diabetogenic (….)”
A: we added the term "obesogenic" in the legend of the table 1
Reviewer 2 Report
In this narrative review, the authors describe the role of endocrine disrupting chemicals in the pathophysiology of diabetes and metabolic disorders. The paper is well written, interesting and exhaustive. The figures are clear and nice.
Author Response
We thank the reviewer for his/her comments.
Reviewer 3 Report
- The manuscript lacks an in-depth discussion of in vitro and in vivo studies that have been conducted on different endocrine disrupting chemicals (EDCs). The EDCs' mechanisms of action contributing to diabetes and metabolic disorders need to be discussed thoroughly. As an example, there are recent studies that examined the role of Bisphenol-A exposure on adipose tissue and adipogenesis, and these studies were not discussed in detail in the manuscript.
- There are several places in the manuscript where references are missing, for example, lines 70-72 and lines 75-76.
Author Response
We thank the reviewer for his/her comments.
Regarding the paragraph on the nature of POPs, we added references and a link to the Stockholm Convention, as mentioned by another reviewer. We have also clarified some of the names of POPs. We added references to the adverse effects previously reported in the literature (especially about neurotoxicity and cancer development: PMID 36430748 and PMID 34273825).
Regarding the comment on the mechanisms of action via adipose tissue, since the review is more oriented on diabetes, we focused on the adverse effects on insulin signaling and not on the alterations of adipose tissue that can be observed with EDCs. Furthermore, we didn’t want to focus our report on BPA alone, for which numerous reviews are already available in the literature.
If the reviewer give us more details, we will modify our text accordingly to improve its quality and thus be able to publish it in the IJMS journal.
Reviewer 4 Report
The authors performed a narrative review evaluating the effects of endocrine disrupting chemicals in the pathophysiology of diabetes. This is a well-conducted narrative review that provides more insight behind this relationship from different epidemiological data as well as both in vivo and in vitro experiments. However, the manuscript requires significant changes to be made prior to being considered for publication. Please find my comments below:
1. Page 1, lines 30-33: the numbers (both the prevalence and the percentage of people with DM2) provided in the text do not seem to match with the corresponding reference #1. Please update.
2. Page 1, lines 33-37: please change “last” to “latest”. It appears that you used the data from IDF 9thedition; please update your data from the IDF 10th edition, available on https://diabetesatlas.org.
3. Page 2, line 57: please provide the names of the databases (e.g., PubMed, MEDLINE, etc.) as well as the keywords used for your literature search.
4. Page 2, line 69: please specify the full compound name for “TCDD”. The listed persistent organic pollutants represent only a fraction of known pollutants. I recommend providing the readers with all the known POPs as listed in the Stockholm Convention.
5. Page 2, line 79: there is an interesting cross-sectional study (PMID 26752053) that evaluated the relationship between insulin resistance/abdominal obesity and dioxin exposure. The addition of this study would complement your manuscript.
6. Page 3, line 141: please provide the appropriate reference.
7. Page 4, line 175: please consider adding a meta-analysis evaluating the relationship between phthalates and insulin resistance. (PMID: 30734259)
8. Page 4, lines 186-7: this is very confusing for the readers to interpret, please rephrase.
9. Page 4, line 197: please provide more insight as to how individual differences in metabolism, critical developmental periods, genetic polymorphisms, etc. affect the bioavailability and effects of EDC.
10. Page 7, line 318: please provide a description with pertinent abbreviations below your Figure 1. You did point out several in vivo effects of different EDC related to diabetes occurrence and later in the text you discuss the effects of immune response in mice after EDC exposure. I recommend you include the diabetogenic potential of EDC from an immunomodulatory perspective in your Figure 1.
11. Page 10, line 423: please provide the abbreviations in your Figure 2 description.
Author Response
First of all, we would thank the reviewer for his/her precise comments. Please find below our answers and the corrections that we performed in our manuscript in order to improve its quality.
Q1: Page 1, lines 30-33: the numbers (both the prevalence and the percentage of people with DM2) provided in the text do not seem to match with the corresponding reference #1. Please update.
A: we updated our results with the new fact sheet of WHO and corrected the numbers
Q2. Page 1, lines 33-37: please change “last” to “latest”. It appears that you used the data from IDF 9thedition; please update your data from the IDF 10thedition, available on https://diabetesatlas.org.
A: we changed the term "last" to "latest" and updated our data with the latest IDF edition. The numbers of people with undiagnosed diabetes is lower than the one we previously reported
Q3. Page 2, line 57: please provide the names of the databases (e.g., PubMed, MEDLINE, etc.) as well as the keywords used for your literature search.
A: we added in the introduction a sentence describing the methodology we used to perform this narrative review
Q4. Page 2, line 69: please specify the full compound name for “TCDD”. The listed persistent organic pollutants represent only a fraction of known pollutants. I recommend providing the readers with all the known POPs as listed in the Stockholm Convention.
A: we specified the full name of TCDD and noted also the reference to Stockholm convention and its annex A and B
Q5. Page 2, line 79: there is an interesting cross-sectional study (PMID 26752053) that evaluated the relationship between insulin resistance/abdominal obesity and dioxin exposure. The addition of this study would complement your manuscript.
A: we added these paper and its interesting data to the paragraph about poisoning event
Q6. Page 3, line 141: please provide the appropriate reference.
A: we let the previous reference only for the adverse effects of BPA on reproductive tract and added the reference PMID 29317319 which concerns the carcinogenic effects of BPA
Q7. Page 4, line 175: please consider adding a meta-analysis evaluating the relationship between phthalates and insulin resistance. (PMID: 30734259)
A: we added this meta-analysis about phthalate exposure and insulinoresistance at the beginning of the paragraph dedicated to phthalates adverse effects
Q8. Page 4, lines 186-7: this is very confusing for the readers to interpret, please rephrase.
A: we rephrased the sentence. Indeed, it was very confusing and the link with triclosan and triclocarban was really unclear.
Q9. Page 4, line 197: please provide more insight as to how individual differences in metabolism, critical developmental periods, genetic polymorphisms, etc. affect the bioavailability and effects of EDC.
A: we rephrased the paragraph about limitations in the interpretation of adverse metabolic effects of EDCs and included these critical points
Q10. Page 7, line 318: please provide a description with pertinent abbreviations below your Figure 1. You did point out several in vivoeffects of different EDC related to diabetes occurrence and later in the text you discuss the effects of immune response in mice after EDC exposure. I recommend you include the diabetogenic potential of EDC from an immunomodulatory perspective in your Figure 1.
A: as suggested by another reviewer, we added the abbreviations used in our figure 1, as the point on immunomodulatory perspective
Q11. Page 10, line 423: please provide the abbreviations in your Figure 2 description.
A: as suggested by another reviewer, we added the abbreviations used in our figure 2.
Round 2
Reviewer 3 Report
The authors addressed the comments.
Reviewer 4 Report
Thank you for providing the answers to my comments. The manuscript can be accepted in present form.